# The Relevance of Building an Appropriate Environment around an Atomic Resolution Transmission Electron Microscope as Prerequisite for Reliable Quantitative Experiments: It Should Be Obvious, but It Is a Subtle Never-Ending Story!

**DOI:** 10.3390/ma16031123

**Published:** 2023-01-28

**Authors:** Antonietta Taurino, Elvio Carlino

**Affiliations:** 1Institute for Microelectronics and Microsystems, National Research Council of Italy (CNR), Via Monteroni, 73100 Lecce, Italy; 2Institute of Crystallography, National Research Council of Italy (CNR), Via Giovanni Amendola 122/O, 70126 Bari, Italy

**Keywords:** electron microscopy infrastructure, laboratory design, environment noises, electromagnetic stray fields, mechanical vibrations, acoustic noise, thermal instabilities

## Abstract

The realization of electron microscopy facilities all over the world has experienced a paramount increase in the last decades. This means huge investments of public and private money due to the high costs of equipment, but also for maintenance and running costs. The proper design of a transmission electron microscopy facility is mandatory to fully use the advanced performances of modern equipment, capable of atomic resolution imaging and spectroscopies, and it is a prerequisite to conceive new methodologies for future advances of the knowledge. Nonetheless, even today, in too many cases around the world, the realization of the environment hosting the equipment is not appropriate and negatively influences the scientific quality of the results during the life of the infrastructure, practically vanishing the investment made. In this study, the key issues related to the realization of an advanced electron microscopy infrastructure are analyzed based on personal experience of more than thirty years, and on the literature.

## 1. Introduction

Over many decades, transmission electron microscopy (TEM) has enabled the study of matter at the highest spatial resolution and accuracy, producing advances in many fields of knowledge, such as physics, chemistry, crystallography, materials science, biology, medicine, etc. Its contributions have so far been directly acknowledged by three Nobel Prizes; in 1971 to Dennis Gabor [1], in 1986 to Ernst Ruska [2], and in 2017 to Jacques Dubochet, Joachim Frank, and Richard Henderson [3], but several other Nobel Prizes have been achieved thanks to electron microscopy. The continuous success of TEM and Scanning TEM (STEM) is due to the constant development of equipment and methodologies capable of answering open questions in a variety of fields of knowledge, and to unveil new routes for understanding nature [4,5,6,7,8,9,10,11,12,13,14,15,16,17,18].

The results attainable from a transmission electron microscope or a scanning transmission electron microscope not only depend on the equipment features but require mandatory laboratory environments in terms of low disturbance, coming from mechanical vibrations, thermal instability, and external electromagnetic stray fields. The stability requirements have become even more stringent with the relatively recent introduction of aberration correctors and high coherence electron sources, which have pushed the spatial resolution of TEM/STEM imaging and spectroscopy experiments, on state-of-the-art equipment, to the sub-Ångström scale [19]. Peter Hawkes reported a moving statement of Ondrej Krivanek: “*The continued use of aging design elements has resulted in a situation where today’s highest performance microscopes are so sensitive that the designers of the microscopes’ foundations need to be concerned about the pounding of ocean waves on a shore 30 miles distant … Even with these precautions, they remain sensitive to adventitious disturbances such as pressure changes due to doors opening and closing, and the low frequency magnetic fields due to passing trucks*” [19]. Nevertheless, even today, there is still not enough consciousness in the scientific community in general, and sometimes even among those committed to electron microscopy, regarding the subtle role that some environmental variables play in the capability to maximize and quantify the information attainable from a TEM/STEM experiment [20]. The role of the laboratory conditions is essential to achieve the theoretical resolution of a TEM, and therefore, the manufacturer requires mandatory stability conditions to guarantee the performances claimed for their equipment. It is worthwhile to note that commissioning tests of new installations always requires, among others, the achievement of guaranteed spatial resolution in the High-Resolution TEM (HRTEM) image [21]. This procedure is, in some way, the result of a standard practice, as up to more than twenty years ago, HRTEM imaging was the approach capable of the highest spatial resolution among the imaging capabilities of TEM equipment [22]. More recently, other methods have been demonstrated to be intrinsically able to achieve higher spatial resolution in TEM-based imaging experiments with respect to HRTEM [23,24]. The advantage of HRTEM resolution tests up to around 25 years ago was evident, as it straightforwardly demonstrated that the equipment was capable of achieving the resolution promised by the manufacturer, providing a signature that the overall quality of the equipment was fair and the environmental laboratory conditions, at least during the commissioning tests, were good enough to achieve the resolution limits guaranteed for the equipment. It is worthwhile to note that this was indeed a guarantee of ultimate performance of the equipment mainly in the past, about until the 1980s. In fact, all microscopists who are old enough to have worked on those microscopes, remember that the achievement of the theoretical resolution at Scherzer’s defocus in HRTEM was not so straightforward and, once reached, it was a reliable benchmark of proper electron optical alignment, equipment operation and environmental stabilities. In more modern instruments, particularly those made since at least the beginning of this century, the level of microscope performances and the assisted capability to align the electron optics very easily, enabled the achievement of lattice images even when everything is not perfectly tuned in terms of the sample and microscope alignment [22,25]. Hence, even though not explicitly stated, the HRTEM resolution test suffers from an historical bias and, today, contrary to the past, the achievement of Scherzer’s resolution does not strictly guarantee that the environmental laboratory conditions are adequate for the best performances of that equipment. Nevertheless, it is too often assumed that, as the TEM reaches the theoretical HRTEM resolution expected for its electron optical setup, this means that it is properly working at the maximum of its possibility. This is not true as evidenced by the fact that nominally identical microscopes can have different performances irrespective of their common capability to reach the theoretical resolution in HRTEM. From this point of view, for example, it would be more significative as overall benchmark, the capability of the equipment to reach the theoretical resolution in STEM High Angle Annular Dark Field (HAADF) imaging, as the latter is much more sensitive to the equipment and environmental stability with respect to the resolution in HRTEM imaging [26]. This is why manufacturers never guarantee the achievement of the theoretical resolution in HAADF, but a resolution worse by a factor two or more, even if the equipment usually achieves better performances with respect to the guaranteed ones. This is not because manufacturers are doubtful about the reproducibility of their equipment, but mostly because they are aware that the theoretical resolution in HAADF is strongly dependent on environmental conditions.

Further misunderstandings regarding microscope performances are due to the attention mostly focused on the attainable resolution and not, for example, on the signal-to-noise ratio, which is obviously fundamental to the real aim of an imaging experiment, namely the possibility to extract accurate numbers from the experiments to model the properties of the specimens. The quality of modern equipment enables much more than the achievement of the HRTEM image resolution at Scherzer’s defocus, and plenty of information is in the corner and new experiments and methodologies can be conceived [27]. Today, most of the limits for advanced TEM/STEM experiments are related to the instability of the environment, which are correctly addressed in the best TEM laboratories in the world [28], but still not in too many laboratories all around the world. Why do many laboratories in the world not pay enough attention to the environmental conditions? Is it a problem of funds or is it a kind of cultural limit, which makes it possible that the search for the best skills of those in charge for the realization of the TEM/STEM research infrastructure is not a mandatory requirement?

In this paper, we will review the state-of-the-art for the realization of laboratory environments capable of maximizing the performances of modern TEM/STEM equipment, by reporting our direct activity in laboratory design and construction, and by referring to successful experiences around the world. The different parameters to be considered for the design of an electron microscopy infrastructure will be considered in detail in the next paragraph, and the solution to minimize the source of disturbance used in some reference-laboratory all over the world will be reported. As a result, it will become evident, in our view, how an electron microscopy infrastructure should be conceived to achieve its ultimate performances in terms of resolution and capability to maximize the signal-to-noise ratio in all of the relevant experiments. This would not only enable the realization of state-of-the-art experiments, but it would open the way for inspiring, conceiving, and developing new methods for the study of the nature.

Nevertheless, we know that too many microscopes are not installed in a proper environment, and we would like also to dedicate a paragraph to our personal experience in laboratory design and construction, over more than 30 years, to mark how, in too many cases, failure in the realization of a TEM/STEM infrastructure could happen simply because the persons in charge of these laboratories did not have the necessary knowledge. In too many cases, the responsibility for the realization of an expensive microscopy infrastructure is left to persons who are not TEM experts, or, even if they have some experience in TEM experiments, they do not have the necessary knowledge and experience for the realization of the infrastructure. It happens too frequently that the equipment is acquired and installed by persons without the necessary knowledge in electron microscopy and, only after the commissioning, the search for people capable of using the equipment starts. Furthermore, operating a microscope and realizing a complex laboratory, designed and optimized for hosting a TEM/STEM infrastructure, do not necessarily require the same skill set. There is a statement in a paper by O’Keefe et al. [20], which is the clue to a rather common way of thinking: “*One potential problem is in interaction with architects and builders. These professionals are very good at supplying things they know about and can guarantee; but it can be difficult to get them to supply or build something with which they are unfamiliar. Good communication is essential to reduce the possibility of anyone making changes without discussion because they “know better”*”. In our experience, engineers with experience in bridge constructions were considered highly qualified for designing and building the laboratory for a high-performance atomic resolution holographic microscope, without interaction with people with adequate knowledge in electron microscopy and in the needs of the infrastructure. The result was, of course, definitely poor. This rather haughty approach was also unfortunately experienced interacting with scientists who managed the realization of other kinds of laboratory or big research infrastructures in their careers, which required a high skillfulness, but they pretended to know what was better for the realization of a TEM infrastructure because they axiomatically “know better”.

Unfortunately, we can only report on this issue; we cannot understand why this happens as, perhaps, this is more a field of study for sociologists or psychiatrists, or both. We believe, according to our experience and thanks to many discussions with colleagues over the last three decades, that this is an issue that needs to be addressed, as it has a strong impact on how funds are spent for the realization of an advanced electron microscopy infrastructure and on the loss of opportunities that a bad realization causes during the lifecycle of the infrastructure itself.

## 2. Historical Survey

One of the earliest scientific texts dealing with laboratory procedures for TEM installation is a volume by R.H. Alderson [29] entitled: “*Design of the Electron Microscope Laboratory*” dating back to 1975. Since then, electron microscopy has experienced many advances and many other articles [20,30,31,32,33,34,35,36,37,38,39,40,41,42] have been published, pointing out new strategies and precautions necessary to accomplish the increasingly stringent requirements of the latest generation of instrumentation. For convenience, in this paper, two tables have been created, the aim of which is to synthesize the results reported in the literature, hopefully providing a comprehensive and readily usable guide to researchers and professionals currently or previously involved in the construction of a TEM laboratory. The tables refer to keynote electron microscopy facilities, realized all over the world by scientists who have decided to share their experience with the microscopist community through dedicated articles, or paragraphs of articles, where the issues related to the preparation of the site and installation of a TEM microscope are thoroughly discussed.

In the following of this paragraph, a historical overview is given, pointing out the advances in terms of instrumental complexity and improved performances, which have impacted on laboratory design and technical solutions for noise reduction.

Since the advent of the first electron microscope, developed by Ruska and his co-workers in the 1930s [43], it became immediately apparent that transmission electron microscopy, although exceeding in principle the performance of optical microscopy in terms of spatial resolution due to the small wavelength of electrons, was well away from the diffraction limit. According to Abbe’s theory [44], the relationship between the resolving power of an optical system and the wavelength of the radiation used to illuminate the object states that the smaller the wavelength, the better the attainable resolution, in principle. The wavelength for an electron accelerated by a potential difference of 75 kV, the value used by the TEM prototype realized by Ruska in 1933, is 4.4 pm, about four orders of magnitude smaller than the wavelength of a photon in the visible range. Nevertheless, in the first observations using an electron microscope, the resolution was of the same order of magnitude as the one achievable using a microscope with an illumination system in the visible light range (Figure 1). The reasons were related to electron lens aberrations, being much higher than the ones typical of an optical system in the visible range, and electron microscope instabilities (high voltage and lens current fluctuations). These phenomena partially covered the deleterious effects of environmental factors, such as mechanical vibrations, thermal instabilities, and external electro-magnetic interference, on the first prototype. Since Ruska’s prototype, the impact of the environment on TEM instruments has grown with the advances in technological devices improving instrumental performances, in particular, the spatial and energy resolution of the electro-optical system and the speed, sensitivity, and resolution of the detectors. In the 1980s, the strategy to improve the spatial resolution followed the route of decreasing the electron wavelength, which meant a higher accelerating voltage, up to more than 1 MV. At the same time, a parallel increase in the stability requirements of the high voltage equipment and of the site was mandatory. As an example, a sub-Ångström information limit was achieved by 1.25 MV instruments, at the Max-Planck-lnstitut für Metallforschung in Stuttgart [30], equipped with a thermionic gun, where special efforts were dedicated to both the site preparation and to the high voltage and objective lens stability (see Figure 1).

It is worthwhile to also mention the case of the National Center for Electron Microscopy at Berkeley, where twin towers were built in 1981 to house the High-Voltage Electron Microscope, operating at 1.5 MV, and the Atomic Resolution Microscope, operating at 1 MV. The United States Department of Energy (DoE) investment for the acquisition of such instruments was about $8 million dollars. In this case, the main issue that the preparation of the site had to face was earthquakes in California. The 30-ton microscope was embedded in 100 tons of cement to lower its center of gravity, floated on air springs, and tethered to a sort of combination pogo-stick and disk-brake foundation, to keep it from walking away in up to 7.8 magnitude quakes [45,46].

In around 10 years, the route for the increase in the acceleration voltage to improve the spatial resolution in an electron microscope was abandoned due to the side effect related to specimen damage. In fact, modern state-of-the-art transmission electron microscopes with advanced aberration correctors are capable of sub-Ångström spatial resolution, both in coherent and incoherent imaging, at an accelerating voltage in a range typically between 60–300 kV.

In this frame, a turning point was represented by the advent of field emission guns (FEG), offering the possibility to focus high electron currents on a sub-nanometric probe [47] and thus allowing both coherent and incoherent imaging in TEM/STEM instruments at improved resolution. This new type of electron gun required additional attention to environment control to guarantee the stability of the probe on the sample. At the end of the 1990s, some of the earlier papers discussing the impact of the environment on modern FEG-TEM instruments appeared, signed by groups at the University of Sheffield and at National Centre for Electron Microscopy (NCEM) of Lawrence Berkeley National Laboratory (LBNL) [31,32].

Another important piece of equipment improving the performance of TEM/STEM instruments and requiring additional attention to environmental sources of noise was represented by aberration correctors. Since the dawn of electron optics, the awareness that spherical and chromatic aberrations could not be corrected by proper lens design, such as in the case of glass lenses, pushed scientists to find alternative strategies, as theorized by the work of Otto Scherzer in 1947 [48]. Early attempts to correct spherical and chromatic aberrations, mainly by the groups of Scherzer and Rose in Darmstadt and Crewe in Chicago, were unsuccessful for two reasons: the first one was the lack of the necessary computer technology and the second one was the presence of incoherent aberrations related to mechanical, thermal, and electrical instabilities, which could eventually hide the effect of the correction. Only in the 1990s a collaborative project between Max Haider and Harald Rose led to a sextupole corrector able to compensate for the spherical aberration of the objective lens. In 1995, the column of an FEI CM200 instrument (Hillsboro, OR, USA), equipped with a Schottky-emission gun, was extended with a Cs corrector for the first time, resulting in a point resolution improvement from 0.24 nm to 0.21 nm [49]. In the same year, the group of Krivanek corrected the spherical aberration of the illuminating probe in a dedicated STEM instrument (a Vacuum Generators (VG) Ltd., East Grinstead, U.K.) by using a computer-controlled quadrupole–octupole corrector, which lowered the Cs coefficient from 3.5 mm to 0.12 mm [50,51,52]. In 2001, a resolution below 1 Å was reached for the first time with a VG HB 501 STEM, working at 120 kV, enabling the observation of single atoms or groups of few atoms [53].

Nowadays, double Cs corrected TEM/STEM microscopes enable both coherent HREM imaging with sub-ångström resolution and incoherent Z-contrast imaging and chemical mapping by scanning a sub-Ångström and high-brightness (10^8^–10^9^ A/cm^2^ sr) electron probe on the specimen.

For these instruments, the stability (thermal, mechanical, and electronic) of the sample, as well as electron optics and detectors, is crucial for the attainment of reliable results. Additionally, the increased height of the column, due to the presence of the Cs correctors, makes mechanical stability issues even more severe, as will be discussed later.

Environmental instabilities are detrimental to different extents, limiting spatial resolution in imaging and energy resolution in Energy Dispersive X-ray Spectroscopy (EDXS) and Electron Energy Loss Spectroscopy (EELS). Furthermore, the serial mode of image and chemical map recording in STEM entails longer acquisition times in comparison with parallel recording, typical of TEM experiments, amplifying the effects of the mechanical and electronical instabilities on the experiments [54]. During sequential acquisition, these effects typically consist of image drifts and distortions, which require the development of dedicated corrections algorithms [55], whereas, during parallel acquisition, loss of contrast and, therefore, of information, is observed [27,34,37,39]. The recent introduction of high speed, high sensitivity direct conversion detectors [56,57] have improved STEM imaging SNR very much, while reducing the acquisition time; but, at the same time, they made possible the development of a new powerful approach, named 4D-STEM, requiring an even longer acquisition time, dedicated noise reduction algorithms, and much greater care in the laboratory stability design [58,59].

## 3. Sources of Noise and Strategies for Their Mitigation in Designing a TEM Laboratory

### 3.1. Preliminary Remarks

In this section, the environmental factors that influence the performance of a TEM/STEM instrument are reviewed and discussed. All noise sources of interest and relevant mitigation approaches are analyzed in detail.

When quantitatively evaluating the effects of noise on TEM/STEM experiments, there are three main parameters to be considered: spatial resolution, signal amplitude, and signal-to-noise ratio.

As already mentioned, spatial resolution is commonly believed to be the key figure of merit in a TEM; therefore, installation requirements, along with tolerance values for instabilities, are provided by the manufacturer for nominal resolution in HRTEM to be guaranteed. Today, in a standard electron microscopy laboratory, the nominal HRTEM resolutions are commonly achieved, provided that the environmental disturbances are within the limits required by the manufacturers; this is also possible because modern TEM instruments are fabricated with built-in shielding devices, which partially cut external disturbances.

The signal-to-noise ratio is another important aspect, which is certainly strongly influenced by the environmental conditions, but it is more difficult to be quantitatively correlated to them. The signal-to-noise ratio can sometimes be more important than resolution. For example, C. Colliex in “*Capturing the signature of single atoms with the tiny probe of a STEM*“ [27], underlined that the possibility to see individual atoms in STEM experiments was guaranteed by the achievement of the required signal-to-noise ratio and not of the required spatial resolution, since the probe diameter on the specimen in their experiments was 0.5 nm, while the incorporated heavy atoms were separated by more than 1 nm. The signal-to-noise ratio becomes critical, for example, when the experimental conditions require the use of a weak illuminating probe, generating a low signal. A typical case is represented by radiation-sensitive organic or inorganic materials. The possibility to study these kinds of materials by electron microscopy is particularly challenging and many efforts have been dedicated to developing methods capable of characterizing these materials, while overcoming specimen damage [60,61]. An example of great success in this issue is the so-called CryoEM, the development of which has been acknowledged in 2017 with the Nobel Prize in Chemistry [62,63,64,65]. When dealing with radiation-sensitive materials, the result is low signal-to-noise ratio measurements, for two main reasons: on one hand, a low dose and a low dose rate are required to prevent damage; on the other hand, radiation damage is common in materials with small transmission function, i.e., scarcely absorbing and scattering objects, *per se* producing a low contrast with respect to the background of the carbon film covering the TEM grid. Low dose and low contrast represent, for these materials, the main limiting factors for quantitative analyses. External sources of noise, even though not affecting the instrumental resolution, can severely worsen, and even jeopardize, the quality of low dose measurements.

The removal of external sources of noise is crucial for TEM/STEM experiments to exploit the highest instrumental performance and capabilities. In this respect, the role of environmental disturbance is fundamental and requires special care when the site for the equipment installation is designed, and during the entire lifecycle of the equipment through proper maintenance. In general, the issue of achieving reliable experimental results with the highest accuracy and resolution is not limited to TEM laboratories. In scientific metrology, it is well known that when targeting the smallest uncertainties, the accuracy of the experiment is limited by the properties of the laboratory room where the experiments are performed. Lassilla et al. [66] pointed out the need for metrology laboratories capable of performing demanding experiments at the highest resolution, on the one hand, and the lack of enough experience and know how to construct high-grade metrology laboratories on the other. This is particularly critical for transmission electron microscopy laboratories. In this respect, in [67], the author reported an impressive table where the environmental requisites of TEM laboratories were compared with those of conventional life science and physical science laboratories, demonstrating how the limits for TEM are orders of magnitude more stringent than for the other laboratories.

In the following, the main sources of noise that require accurate evaluation for their mitigation will be discussed. They can be distinguished according to the following categories:(1)sources of electromagnetic noise;(2)sources of thermal noise;(3)sources of mechanical noise.

One must consider that even the operator of the microscope is a further source of disturbance, as he/she moves, breathes, and emits heat and sounds; it is a common experience, when the operator’s chair has a steel component, that the electron energy loss spectrum shifts as the chair moves. Therefore, in recent, more advanced instruments, the operator is moved in a room away from the microscope and operates the equipment in a remote mode [35,68].

TEM manufacturers always provide each TEM/STEM instrument with a list of minimum requirements to be fulfilled to achieve the guaranteed resolutions and performances. Nevertheless, equipment performances and the quality of the experimental results can go much further, depending on the laboratory design [20,33,69]. Therefore, instruments with the same nominal performance cannot achieve the same results if they work in environments that are not comparable. On the other hand, a comparison of equipment with different features could be meaningless if the environment is not the same. In this respect, for example, two illustrative cases are discussed in the literature: one is a reliable comparison between two different sources of electrons, a Schottky FEG versus a cold FEG, made possible as the electron sources were installed on the same microscope and in the same laboratory, accurately designed to optimize the equipment performance and to screen the external disturbance that otherwise would have hidden the differences between the two electron guns [68]. The complementary example is reported in [33], where an increase of the spatial resolution of a CM 30 FEG microscope (Philips Electron Optics GmbH, Eindhoven, The Netherlands) from 1.2 Å to 0.9 Å was observed when the instrument was moved from the former laboratory, at the Technical University of Dresden, to the newly constructed laboratory, at Triebenberg, directed by Annes Lichte (see Table 1 and Table 2). In [33], Lichte underlined the importance of the site choice, in addition to the “best art of engineering” for the laboratory design and the building. The site, a soviet radar spy station before German reunification, was selected by Lichte for its characteristics of isolation, sat atop a hill away from roads, railways, and airports, free from power lines and even shielded from wind by means of a dense belt of trees; there, the AC stray field was found to be of about 1–2 nanotesla, a factor of a hundred better than usually required by electron microscopy factories for TEM labs. In the article *Electron Holography Lab Pushes Resolution Limit*, Physics Today reported [70]: “*Lichte has recognized that the building is an integral part of the whole instrument. The time is ripe. Instruments and algorithms only recently got to the point that they are limited by buildings. Now he is truly limited by the aberrations of the instruments*”.

Based on the latter considerations, the preliminary action to understand if a site can be designated to the realization of a modern transmission electron microscopy laboratory aims at detecting and quantifying the eventual presence of external sources of disturbance. The main factors to be considered are:(i)proximity to street traffic, railways, et similia.(ii)proximity to noisy laboratories and/or technical rooms, like mechanics, elevator shafts, electrical rooms, server rooms, etc.(iii)proximity to highly populated zones, like halls, meeting rooms, classrooms, canteens, etc.(iv)cross interference between adjacent TEM/STEM laboratories.

A geologic survey is mandatory to inspect the soil structure for the realization of a proper vibration isolation block where the equipment must be positioned.

As the conclusive remarks of this paragraph, we would like to stress the impact of TEM facilities in terms of costs. When building a new TEM facility, a realistic estimation of the total costs must also consider, as a significant part of the investment, the costs for the construction of the laboratory rooms. In the experience of the Brookhaven National Laboratory, at Long Island (Table 1 and Table 2), only two of the four high-accuracy laboratories for aberration-corrected microscopes were initially constructed, due to their unexpectedly high cost [39]. As Lichte declared [70], the cost of the new building hosting the Triebenberg laboratory was roughly the same as two of his electron microscopes. These two examples are indicative of the efforts that could be necessary to achieve a stable environment.

In the following paragraphs, specific sources of noise and the relevant mitigation measures will be discussed with reference to the existing literature, the data of which are schematically reported in Table 1 and Table 2 for convenience. In particular, Table 1 reports the list of the considered electron microscopy facilities, with reference to the institutions where they have been realized (first column); for each facility, details about the construction of the building (second column), the rooms (third column) hosting the microscopes, and their ancillary equipment are reported; in the fourth column of the same table, details regarding the installed microscopes (name, year of installation, technical specifications) are reported, while additional information related to their performances and other technical data pertaining to the instabilities issues affecting the performances themselves are listed in the fifth column. For each TEM laboratory reported in Table 1, Table 2 outlines the limits required by the manufacturers for each type of noise, along with the relevant strategies and remedies adopted during the realization of the TEM laboratory.

### 3.2. Sources of Electromagnetic Noise

The first mandatory action in a site where a TEM/STEM is planned to be installed is to accurately measure the external AC and DC sources of stray fields and the propagation for the mechanical disturbance at specific frequencies that mostly affect the equipment.

As far as electro-magnetic noise is concerned, the equipment itself is a source of noise. Zhu et al. [39] reported that, before microscope installation and operation, the measured induced AC fields in the instrument room, in all x, y, and z directions at different heights, were below 0.005 mG, whereas after the instrument and the ancillary equipment were switched on, the average AC magnetic fields at 60 Hz increased to 0.15 mG in the z-direction and 0.08 mG in x-y directions, which means an increase by a factor 30 in the z-direction and by a factor 15 in the x-y plane. Please note that, even though not explicitly stated in reference [39], the values reported in that paper are supposed to be rms and not peak-to-peak as commonly used for AC fields.

The effects of EM noise are detrimental in many respects, as they produce scanning distortions in STEM images, aberrations in high-resolution TEM images, and loss of energy resolution in EELS experiments.

In [38], ad hoc experiments were planned and performed by the authors to measure the effects of EM noise in typical TEM/STEM experiments. By using a reference layer of SrTiO_3_, grown on silicon, 5-unit cells thick (1.96 nm), they measured the distortion on STEM images, induced by an external AC field, generated by the current circulating in a coil one meter in diameter. Sensitivity factors to 0.5 Å/mG and 1.42 Å/mG were measured for a Tecnai F20 with a monochromator TEM/STEM instrument and for a VGHB501A UHV dedicated STEM, respectively.

Typical sources of AC fields are represented by currents lost towards ground, due to bad ground connections. These currents flow through metal conduits in the microscope room, causing the generation of stray fields [34]. These problems can be easily fixed but are difficult to isolate. Therefore, a simple way to overcome them is to remove all old wiring and to enclose new cables into electrical trenches far from the microscope, which, in addition to providing electromagnetic shielding and eliminating hazardous obstacles on the floor, prevents dust accumulation and facilitates floor cleaning. The floor tiles must be conductive and grounded to prevent electrostatic charging. Special attention must also be paid to lighting; for example, dimmable incandescent lighting must be chosen to eliminate the radio-frequency noise due to the electronics of fluorescent lighting, the use of which must be limited to maintenance operations [39].

Additionally, quasi-DC fields may be generated by metal objects moving close to the microscope, which are responsible for energy shifts in the alignment of the EELS spectrometer, making the interpretation of the spectra unreliable. A shift of the order of 1 eV in the EELS spectrum can be caused by moving the iron wheels of an office chair; therefore, wooden chairs are better suited for TEM laboratories (see Figure 2). In [34], the authors reported on a series of typical moving objects which could cause these types of problems.

A simple way to mitigate the effect of EM fields, and the eventual environmental thermal fluctuation, is to host the microscope in a large room, since the intensity of EM fields decay very rapidly with distance, and the large volume of the room behaves as a thermal buffer stabilizer.

Screening strategies for electromagnetic disturbances can be passive or active: passive shielding uses high permeability metals or metal-alloys, which are commercially available in sheets or foils of different size and thickness. In Figure 3, the attenuation efficacy (in $dB=10log\frac{B_{unshielded}}{B_{shielded}}$, where *B_unshielded_* and *B**_shielded_* are the intensities of the field before and after attenuation, respectively) of three possible candidate materials for EM shielding is shown.

Ferromagnetic alloys exhibit good behavior since their attenuation slightly drops as frequency decreases in comparison with pure metals, such as aluminum or copper.

Typical examples of branded ferromagnetic alloys include MuMetal^®^, Netic^®^, Finemet^®^ and Metglas^®^, to name a few. Among these, Mu-Metal, a nickel–iron soft ferromagnetic alloy with high nickel content (80–82%), is particularly appreciated [34,38] for its attenuating properties and for its availability in a wide range of stock thicknesses from 0.36 mm to 5 mm.

The required thickness of the shielding foil is related to the skin depth, given by $\delta =\sqrt{\frac{2}{\sigma \mu \omega }}$, where *σ* is the conductivity, *µ* the permeability, and *w* the frequency. The formula suggests that low frequency fields are difficult to attenuate; therefore, thicker and more expensive foils are required in this case.

Active cancellation systems consist of Helmholtz coils, running around the microscope room, and feedback wideband (including DC) sensors, which measure the magnetic field to be canceled. These systems are very effective in canceling the field at an exact point and at high frequencies, where the feedback sensor is more sensitive. They are not so efficient for inhomogeneous stray fields or stray fields produced by close sources. For example, in a small room, if the field is canceled at the gun level, it would be enhanced at the spectrometer [38]. More efficient systems, based on triaxial magnetic field compensation, have been designed for the new electron microscope at the Graz Centre for Electron Microscopy [71], by optimizing the position of the sensor and the shape of the coil, both tailored to the room geometry and interfering fields. From personal experience, sometimes it happens that strange blurring can be experienced during imaging or spectroscopic experiments. After sometimes long hunting for the noise source, it disappears by resetting the active compensation system. In Table 2, the strategies used by different TEM facilities to attenuate EM stray fields are reported, along with the limits required by the installation specifications and the issues experienced during the laboratory realization.

### 3.3. Sources of Thermal Noise

One of the most important requisites for TEM equipment is to maintain the surrounding environment at a fixed (around 20 °C) and stable (±0.1 °C) temperature. Temperature variations cause a drift of the specimen, of the microscope electronics, and of the mechanical tolerances in components, including microscope lenses, detectors, aberration correctors, and scan coils.

As in the case of the EM fields, the size of the room plays an important role in temperature stability, since larger spaces around the microscope will better dampen any heat spike within the laboratory. Two types of temperature control systems are commonly used, i.e., forced air systems and radiant panels; they remove heat and keep the temperature as required. Forced air systems remove heat by convection and conduction. Since the heat capacity of air is very low, large airflows are necessary to remove heat. Airflows cause air pressure on the microscope column, resulting in mechanical vibrations (see next paragraph). This is one of the reasons why, in modern laboratories, heat removal by air flow is minimized by using two more convenient strategies: the first one is passive and consists of locating all heat-generating equipment that can be separated from the microscope, such as power distribution racks in a service room, separated from the microscope room. The second one is active and is based on radiation, instead of conduction and convection, as the main mechanism for heat load reduction, by using thermal masses placed in the microscope room. Radiant panels [72,73] are installed on the room walls and chilled water is circulated through them; they can regulate the temperature to better than 0.1 °C, and, in case of thermal drifts, for example due to the entrance of a person in the microscope room, the return to equilibrium is quite rapid.

Therefore, forced air systems are minimally used to regulate the temperature and mainly to control humidity, thus avoiding water condensation on the radiant panels and on the cooled parts of the microscope (electronics, pumps).

The primary effect of using radiant panels instead of a forced air system is reduced drift of image and spectra, helpful for long acquisition times, like during the frame integration of STEM image acquisition, 4D-STEM, or during analytical experiments such as EELS and EDXS chemical mapping. Secondly, since both spectrometer and high-tension supply are sensitive to temperature changes, higher temperature stability results in more reliable spectra as a function of time.

### 3.4. Sources of Mechanical Noise

Mechanical instabilities are crucial, especially for the microscopes of the latest generation, since the presence of aberration correctors and/or monochromators makes the column longer than in old microscopes and more sensitive to mechanical vibrations. The stiffness of the microscope’s column linearly worsens with height and roughly improves with the second power of diameter. Therefore, modern microscopes have been completely redesigned by manufacturers to achieve better stability. Nevertheless, environmental vibrations remain an important source of instabilities, especially at the level of the gun and the specimen. The gun is placed on top of the column and is, therefore, subjected to maximum sway, with a detrimental effect on probe formation. Stage movements are also negative for high resolution imaging and spectroscopy. Vertical vibration of the specimen within the objective lens pole pieces results in a spread of focus of the image, which limits the attainable resolution. Horizontal vibration, usually more in one direction, will smear out the image, also limiting the resolution.

One of the main difficulties related to the treatment of vibrations is the microscope’s sensitivity to low frequency vibrations, in the range of a few Hertz; these are the most difficult to eliminate from the microscope’s environment.

There are different sources of mechanical noise; they can be distinguished because of the medium responsible for the noise propagation, i.e., air and soil.

#### 3.4.1. Mechanical Noise from Air: Thermal and Acoustic Sources

Mechanical noise from air is related to air movements, mainly generated by acoustic waves and thermal gradients, deriving from temperature control systems.

To estimate the air pressure that an acoustic source can produce, let us consider the case of a very quiet ambient, which means an acoustic level of about 40 dB. The threshold for annoying acoustic noise is conventionally set to 70 dB, which means it is increased by a factor of one thousand with respect 40 dB. Normal conversations are around 55 dB; 30 dB when softly whispering; door slamming is 90 dB, decreasing to 50 dB for gently closing, and footsteps are a maximum 63 dB. A sound of 40 dB corresponds to an air pressure of 2 mPa, being the sound pressure level *L**_p_*** (in dB), a logarithmic measure of the sound pressure *P*, according to the equation L_p_ = 20 log_10_ (P/P_0_), where P is the root mean square sound pressure and P_0_ is the reference sound pressure. The commonly used reference is the sound pressure in air for which the value is 20 µPa (considered as the threshold of human hearing). A 40 dB noise produces an air pressure of 2 mPa, which acts on the microscope column, having a section of about (0.3 m × 3 m) = 0.9 m^2^, with a force of 1.8 mN. With a resonance frequency of a few Hz and a mass of about 1000 kg, the spring constant of the microscope is k = mw^2^~10^6^ N/m, which means a displacement, from Hook’s law, of about 2 nm. This rough calculation demonstrates how a solid and heavy object, such as a TEM microscope, can oscillate under the effect of weak air perturbation; this would not produce visible effects if the TEM holder is fully integral with the column; unfortunately, this is not completely true because a simple test, like clapping hands or talking near the column, will immediately show visible high magnification image shaking. In [34], using a high-sensitivity barometer, the authors measured a deflection of their side entry rod of 0.1 nm/Pa. Therefore, to protect the sample holder from common pressure variations, due, for example, to weather changes or pressure waves caused by a door opening, some years ago the Jeol 2010F FEG STEM were optionally retrofitted with an airtight airlock cover (a “clamshell”), purposefully designed and built by the manufacturer. Today, such precautions and their evolutions have been adopted on all high-level TEM instruments.

Another subtle source of pressure variation on a typical side entry specimen holder is due to the extraction air fan. The high-tension tank of a microscope usually requires, for accelerating a voltage higher than 200 kV, to be filled by a gas with a high dielectric constant to avoid discharge within the tank. Usually, SF_6_ gas is used for this purpose. SF_6_ is heavier than air, and hence, for safety reasons, a fan extractor at floor level is mandatory for the unlikely events of SF_6_ leakage. The extractor is usually monitored by a gas sensor and must be directly connected to the outside. The external hose must be properly screened to prevent possible extraction speed variations in the presence of external strong wind variations. This is particularly important when the laboratory is placed in areas with frequent strong winds, as we personally experienced around Trieste, famous for the Bora, a wind characterized by sudden and strong intensity variations. The Bora was found to directly affect TEM experiments, and this was unexpected at the beginning, and not so trivial as to be understood and fixed.

The temperature control systems must be designed and realized in such a way that air movements in the laboratory are minimized. Forced air systems should diffuse air as much as possible to produce laminar flow. To this aim, a technical solution is represented by a laminar ceiling made of perforated panels, which provide downward and even air distribution. This solution is quite expensive and difficult to retrofit. Another solution, much cheaper and particularly suitable for retrofit, makes use of a duct sock connected to the air inlet; the natural wave of the sock tissue lets air softly diffuse out, as reported in [38]. In [34], the authors created a home-made simple but effective test which could be easily carried out to roughly evaluate the airflow; the so-called “toilet-paper test”: 12 × 0.25 inches. Strips of single-ply paper are attached around the microscope, and, if they deflect at the bottom by more than an inch, then the airflow exceeds 20 ft/min. At the time the article was written, 15 ft/min was considered acceptable for a 0.2 nm resolution in STEM, but, today, the sub-angstrom resolution achievable with the modern state-of-the-art instrument requires much lower limits, as reported in Table 2, and highly sensitive airflow detectors for their quantitative measurements. Nevertheless, the test can be performed as a preliminary test to assess the condition of the room, or to periodically monitor time stability.

Air movement caused by acoustic waves impacts microscope performance, depending on intensity and frequency. Common sources of acoustic noise are computers, power racks, pumps, chillers, electronics, and air inlets, but external sources can also contribute. Noise attenuation can be achieved by using acoustic shielding, and since common sound damping materials like polyurethane or other foams are inefficacious at low frequency (f < 130 Hz) where the microscope is more sensitive, an effective solution for low frequency attenuation is to use fiberglass absorbers, placed in front of the laboratory walls with an air gap in between. Nevertheless, as in the case of the EM stray fields, before applying any attenuation strategies, the most reasonable and effective approach is to identify the sources of acoustic noise and remove them, or put them as far from the microscope as possible. For example, all ancillary equipment, such as power racks, pumps, and chillers, must be isolated in a separate service room [20,39]. In [37], the presence of noise, due to acoustic coupling between the turbomolecular pumps and the column, was detected through the appearance of high frequency reflections in the FFT of high-resolution STEM images. The problem was only completely solved by the manufacturer, who modified the original vacuum system layout of the instrument, as all isolation attempts were demonstrated to be ineffective in canceling the noise, only in attenuating it.

For acoustic noise, it is difficult to define limits and thresholds, since each microscope reacts to acoustic waves depending on its own resonances. Generally, an accredited criterium is to reduce the sound intensity below 40 dB [39], as reported in Table 2.

#### 3.4.2. Mechanical Noise from Soil

Mechanical vibrations from the floor supporting the microscope may have different origins: one is related to microquakes, caused by movements of the Earth’s crust, sea waves, mountains, and even glaciers; these “microseisms” contribute to a background noise which cannot be eliminated. In [35], the authors showed plots of the micro-seismic activity near NCEM at LBNL in Berkeley recorded during a Pacific storm on 25 December 1996.

Local sources are road or rail traffic, heavy machinery, and similar items which produce vibrations in the soil, the propagation of which can reach the bedrock under the microscope. Vibrations in the low-frequency range (<5 Hz) are the most critical for the microscope; their attenuation can be efficiently achieved using large masses, i.e., by placing the microscope on a concreate slab with suitable sizes that must be tailored, after a geological inspection, as reported in Section 3.1. To correctly dampen vibrations coming from the soil, the slab weight must be generally tens of times larger than the microscope weight; the slab must be also isolated from the remaining floor with a few cm-wide trench, avoiding the transmission of movements of the building hosting the microscope laboratory. In Figure 4, a scheme depicting the main characteristics of the isolation slab is reported.

In addition to the use of large slabs as a passive measure to attenuate vibrations, all microscopes have their own passive air cushion (or springs) isolation systems that provide enough isolation for frequencies above 10 Hz. Furthermore, active systems are also available, which can actively compensate for disturbances in a wide range of frequencies, even in the critical range 1–5 Hz. As reported for the active electromagnetic compensator, a malfunctioning of these devices could also happen and could require a reset of the device to restore proper operation.

As for the other sources of noise, prevention, when possible, is always more effective and less expensive than mitigation. Therefore, an a priori evaluation of the possible sources of vibrations should guide the choice of site for the infrastructure to be constructed ad hoc. When possible, closeness to street traffic, elevators, and even highly frequented corridors must be avoided, as even foot fall impact must be mitigated for. In Table 1 and Table 2, the technical strategies applied at different facilities to limit mechanical noise from soil are reported.

## 4. Learning from Some Direct Experiences

We were, for the first time, committed to the realization of a TEM/STEM laboratory, more than 30 years ago. At that time, we experienced some problems related to the poor skillfulness of the people committed to the project and we saw the relevant heavy influence on the performances of the equipment installed. We also experienced some of these problems during the realization of another TEM/STEM laboratory at the beginning of 2000, and again in another, to which we were committed very recently. It seems that some human-related problems are time invariant.

During the construction of the first laboratories, there was evidence of how apparently small differences were definitely important in the performance of expensive state-of-art microscopes throughout their entire productive lifecycle. At that time, three brand new state-of-art TEM microscopes were installed in a period of about two years, at the Centro Nazionale per la Ricerca e lo Sviluppo dei Materiali (CNRSM) in Italy. At that time, those laboratories represented the largest electron microscopy infrastructure in Italy. The microscopes were provided by three different manufacturers. An expert group of researchers, with significant experience in TEM, designed the general layout of the laboratories and planned that all the microscopes would be installed on identical, individual (3 × 3 × 3) m^3^ anti-vibrating blocks, located within an air-conditioned (16 × 12) m^2^ large room, with a height of 7 m. Furthermore, around each microscope, individual boxes were planned with dedicated air conditioning and low environmental disturbance in terms of acoustic noise, mechanical vibrations, electromagnetic stray fields, etc. The features of each box were not decided in the general layout plan, and were left to the persons in charge for each TEM installation. As a result, the sizes and constructive materials of the individual boxes were different from equipment to equipment, not because the equipment were different, but because the persons in charge had different expertise and points of view. Two of the involved microscopes had similar types of condenser illumination systems, known as three condenser lens illumination system [74]; whereas the third one had a Köhler condenser illumination system [74], and largely different resolution capabilities related to the objective lens used. This is why a comparison of the latter with the other two microscopes is not straightforward, and is perhaps meaningless for our purposes. In fact, we think that the example of the installation of the two transmission electron microscopes with a similar illumination system could be more helpful in understanding the importance of apparently small elements in the construction of the boxes containing the microscopes.

For one of the microscopes, the first one installed, the design of the box, the materials used, and the choice of the low vibration water-chiller dedicated to refrigerating the equipment was performed by a group of expert electron microscopists in daily collaboration with the engineers committed by the manufacturer for the installation and control of the environmental conditions around the microscope. We choose not to report on the brand of the microscope, as here, we focus on the influence of the environment on the equipment and not on the brand of the microscope, which is not relevant to this purpose, and we will refer to this equipment as microscope #1. Microscope #1 had a LaB_6_ cathode, a maximum accelerating voltage for the primary electrons of 300 kV, a side entry specimen holder, and an electron optical system, giving a theoretical interpretable resolution at Scherzer defocus, at 300 KV of accelerating voltage, of 0.23 nm. At that time, the class of these kinds of microscopes was called “analytical”, as the design of the objective lens aimed at having enough space between the pole pieces to enable a relatively wide range of tilts of the specimen holder along two orthogonal directions, and being suitable to host one or more energy dispersive detectors to analyze the x-ray emitted by the specimen under electron irradiation (EDXS) for analytical purposes [22]. A wider gap in the pole pieces made this configuration less performant in terms of HRTEM image resolution, the latter being directly related to the spherical aberration of the objective lens [74]. Microscope #1 was placed in a 4 × 9 m^2^ large box, with a height of 4 m. The walls of the box consisted of two Al foils, about 1-cm thick each, with a polymeric thermal insulating layer, about 15-cm thick, in between. The size of the room and the materials used for its construction made the box very robust with respect to external acoustic noise, external electromagnetic stray fields, and external thermal variation [20], as discussed in detail in the previous paragraph. All of the noisy ancillary equipment, including the water-water type chiller, were placed outside of the box. The air conditioning within the box was realized by using dedicated equipment, placed in an open space away from the laboratory, and the cold air was directed between the top of the ceiling and a drop (or suspended) ceiling made of micro-pored panels, so that the cold air descended in the room from the top without turbulence on the microscope column. At that time, radiative panels to stabilize the thermal conditions were not so popular. The care and precision in the realization of the TEM Laboratory largely fulfilled the manufacturer’s requirements for the microscope’s installation. The commissioning tests straightforwardly achieved the resolution benchmarks declared by the manufacturer. The stability of the laboratory environment straightforwardly produced lattice imaging with the expected spatial resolution, but also with a very good signal-to-noise ratio (SNR), enabling a proper setup of the illumination conditions and of the image acquisition both on photographic film (still largely used at that time) and on the Slow Scan Coupled Charged Device (SSCCD) camera. The SNR of the HRTEM images is still not officially recognized today in the acceptance tests for new TEM installations, but it is fundamental to maximize and to quantify the image contrast, deriving detailed sample information otherwise not accessible [27], and enabling the development of new methodologies. As an example, the environmental stability around microscope #1 and the relevant equipment performances enabled innovative quantitative measurements of the concentration profiles of ternary semiconducting III-V alloys by HRTEM imaging [75]. The role of SNR is of particular concern today for atomic resolution studies of single particles of radiation-sensitive organic and inorganic materials, which, with proper equipment setup, electron dose, and electron dose rate, disclose the access to the atomic structure of radiation-sensitive nanoparticles [60]. The second microscope, named microscope #2, was especially designed by the manufacturer for HRTEM imaging and was, at that time, the medium voltage equipment with the better lattice imaging resolution available on the market. At that time, this kind of microscope was extremally stable from a mechanical and electrical perspective, and the objective lens was designed to reach the best spatial resolution performances without spherical aberration correctors. Microscope #2 was equipped with a LaB_6_ cathode, it had 400 kV as the maximum acceleration voltage, a very stable top-entry specimen holder [74], and electron optics, with an objective lens with a small gap and small spherical aberration coefficient that enabled an interpretable spatial resolution at Scherzer’s defocus of 0.16 nm. The performances of this microscope were expected to be particularly striking when the highest HRTEM imaging resolution was necessary to understand the properties of a specimen. Unfortunately, the features of the construction of the box for the installation and the features of the ancillary equipment were decided by the senior scientist in charge that had no direct experience in TEM. He was supported by personnel with some experience in TEM experiments, but no experience on how a TEM laboratory should be built. They decided to reduce the cost of the construction of the box around the microscope, making it smaller than the previous one. The box was hardly large enough to contain microscope #2, and had thin walls. The chiller to refrigerate the water flowing in the microscope lenses and the electronics was of the air-water type, which is a bit less expensive with respect to the water-water type, but is extremally noisy when the air fan is on. Consequently, it was placed very far away from the microscope box in the basement below the microscope room. This choice resulted in a slightly turbulent water flux in the lenses. For the conditioning, the size of the ceiling area, about half of that of the box of microscope #1, required a higher air flux through the pores, resulting in a bit of air turbulence on the microscope column. These relatively small perturbations in the environment did not prevent the success of the resolution test for the commissioning simply because, to reach the resolution for the commissioning, the air conditioning was switched off for the short time necessary to acquire the images, but affected all of the experiments made over the years on microscope #2. The HRTEM experiments made on microscope #1 always had a much better SNR with respect to the experiments made on microscope #2, on the same TEM specimens, for all the time in which these microscopes were operated. The performances of microscope #2 were largely inferior with respect to the other microscopes of the same type installed in well-designed laboratories in Europe at that time, where we performed HRTEM experiments on the same specimens used on microscope #2. The savings for the box, the chiller, the air conditioning, etc., for microscope #2 was around less than 1% of the cost of the microscope, which corresponds to the expenses necessary for operating a microscope of that kind for a few months, but affected the results and the utility of microscope #2 forever. From this example, it immediately emerges that the few thousand euros saved during the equipment installation did not justify the loss of scientific opportunities due to inaccurate installation, and produced a much bigger economic loss, as a microscope of that type drains thousands of euros for electricity, water, maintenance, etc. The problem was clearly only due to the inadequacy of the personnel in charge of the installation of the equipment.

The second example of building a TEM laboratory to which we were committed dates to the beginning of this century, and we believe that it can provide good advice on how a good laboratory design can positively influence the productive life of a TEM/STEM and open new routes for novel approaches. The realization of the laboratory was directed by one of the authors, and we also received several demands from people who were not experts in TEM and in the realization of TEM/STEM infrastructures, but who were strongly convinced that it was not necessary to have direct particular experience in this field if someone had strong experience in realizing other kinds of research infrastructure, such as Synchrotrons, Synchrotron beamlines, different kinds of laser spectroscopies, etc. Fortunately, these pressures did not produce problems, but only a bit of disturbance and the construction of the laboratory was realized considering all of the direct positive experiences made in the installations of microscope #1, and the negative experiences in the installation of microscope #2, but, of course, also considering the information available in the literature, and by receiving the valuable suggestions of colleagues with significant and demonstrated experience in the realization of TEM/STEM infrastructures in Europe, and open-minded colleagues from other disciplines of physics. As a result, the laboratory environment was extremally stable, more than what was required by the manufacturer of the JEM 2010F UHR TEM/STEM installed, and guaranteed, for more than twenty years, state-of-the-art experiments. As far as we know, this was the only instrument of this type in the world that reached the theoretical resolution in high angle annular dark field (HAADF) imaging of 0.126 nm [69]. The thermal, mechanical, and electromagnetic stability of the environment enabled the realization of experiments otherwise not possible. Herein, we report a few examples. It was possible, for the first time, to measure at atomic resolution the distribution of a nominal monoatomic delta doping in a host matrix by HAADF [76], and to conceive and to develop new methods to quantify the distribution of the species in multilayered specimens [77,78]. The equipment produced high SNR images in HAADF and HRTEM, but also, thanks to the thermal stability, it was possible to acquire EDXS spectra from the interface of heterostructures, achieving cross-correlated information from different imaging methods and EDXS [79]. This allowed us to solve a long-term problem concerning the chemistry and the structure of a few atomic layer phases in ZnSe/GaAs heterostructures capable of modifying, to several orders of magnitude, the dislocation density in the ZnSe epilayer and changing, in orders of magnitude, the lifetime of the laser devices based on ZnSe/GaAs heterostructures [80]. The environmental stability, which was much better than that required for a Jeol 2010 F UHR microscope with a interpretable resolution at Scherzer defocus of 0.19 nm, enabled us also to conceive and to demonstrate new imaging methods capable of improving, by a factor three, the spatial resolution of the equipment. In fact, we developed a coherent electron diffraction imaging method, and we imaged the structure of single nanoparticles [24] and extended specimens [81] with a resolution down to 70 pm, which is, so far, the best results achieved in the world using these approaches. This equipment also enabled the development of a new method, called HoloTEM, to study radiation-sensitive materials by coupling in-line holography and low-dose HRTEM methods, where a short as possible exposure time is mandatory to reduce the electron dose on the specimen [60]; whereas high contrast low dose approaches are mandatory due to the low image contrast achievable on materials made by low atomic number atoms.

We were recently committed to the realization of a laboratory where a state-of-the-art TEM/STEM microscope, equipped with a high coherence cold field emission emitter, double correctors for spherical aberrations, powerful systems for EDXS, an electron energy loss spectrometer for spectroscopy, and energy-filtered imaging and diffractions, a state-of-the-art direct conversion detector, etc., which costs several millions of euros, will be installed. Despite the evidence of the strong advances achievable by properly designing a TEM/STEM laboratory and the need of true skill to design such a facility, we still faced the uncomfortable problem of people who are, in some way, in charge of the facility realization, having experience in other field of physics and not in electron microscopy, and less than that in the realization of an advanced TEM/STEM infrastructure. We are aware that this experience is unfortunately more common than it should be, and, irrespective of the latitude, it can happen in many places in the world, where things are done by someone simply because they think that they know how to do better [20]. The reason for that remains, for us, incomprehensible. The only thing we could say to those who are planning a TEM/STEM facility is to leave this job to persons who really have experience in this matter. It seems obvious, but history has taught us that, evidently, it is not obvious at all.

## 5. Conclusions

The setup of a TEM infrastructure is a task that requires significant experience, not only in TEM, but in this specific issue, and cannot be left to non-expert microscopists or, even worse, to those that have experience in other fields and think that this is enough to properly realize a TEM infrastructure.

Here, we provided personal experience and the relevant state-of-the-art knowledge in the literature to tackle the realization of a laboratory capable of maximizing the capability of state-of-the-art microscopes to study the matter at the highest spatial resolution and accuracy, establishing the operative conditions to stimulate the development of new TEM/STEM-based methodologies necessary to further advance the knowledge.

The realization of two tables, which summarize in all of the information scattered in the literature on this subject, provides a useful guide, readily available for consultation, to all researchers and professionals who aim at building up a new and properly working TEM laboratory.

## Figures and Tables

**Figure 1 materials-16-01123-f001:**
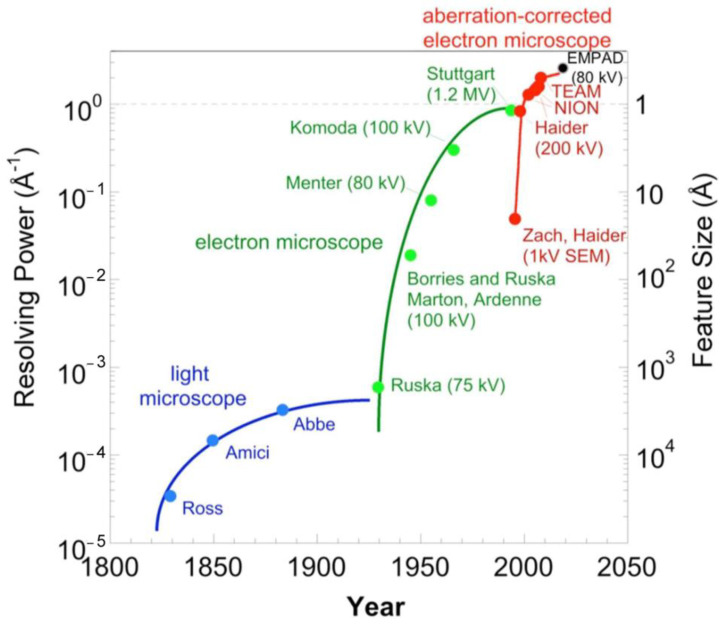
Historical evolution of the electron microscope resolving power (reproduced from: “*Smaller than the space between atoms: The technology behind the highest-resolution microscope image*”, courtesy of Muller, D.A.).

**Figure 2 materials-16-01123-f002:**
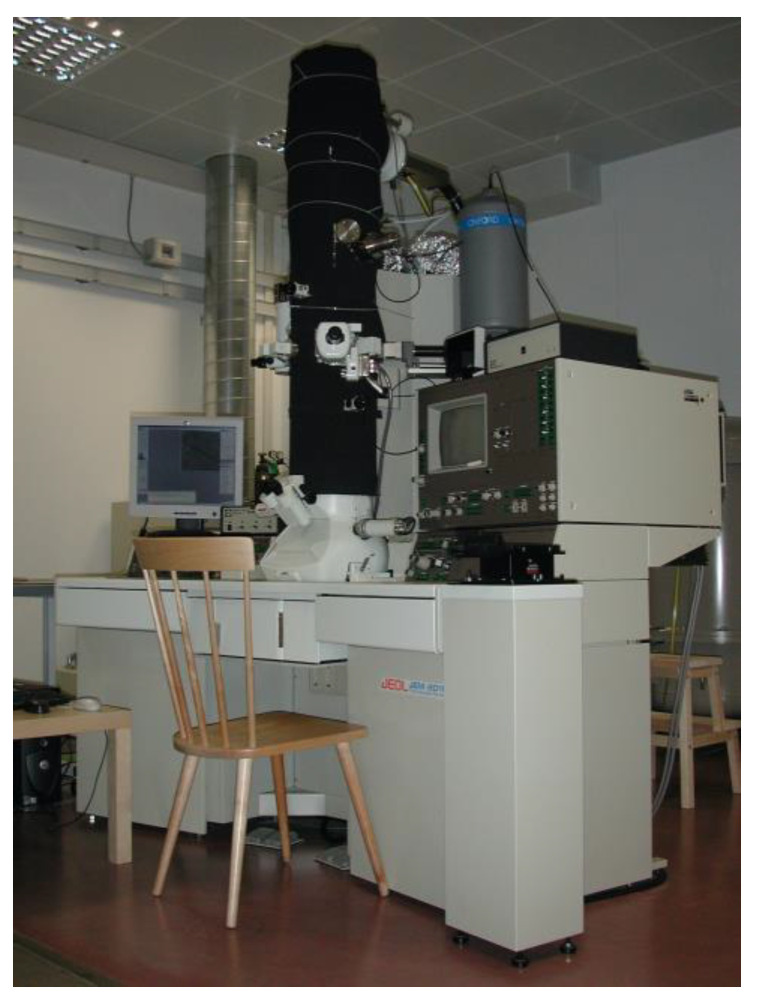
Photograph of one TEM/STEM laboratory at the ‘Istituto Officina dei Materiali’ of the Italian National Research Council (IOM CNR) in Trieste, showing a classic colonial wooden chair, which was used by the operator to prevent magnetic noise during the experiments.

**Figure 3 materials-16-01123-f003:**
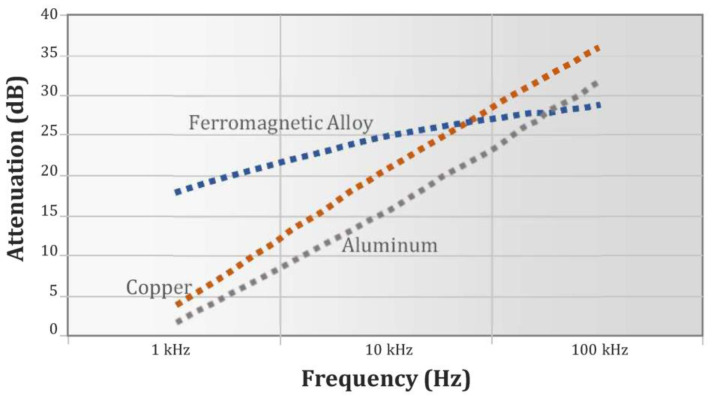
Attenuation performance of metal materials commonly used in EM shielding. Source: Ott, H.W. Noise reduction techniques in electronic systems, John Wiley and Sons. New York, 1976 (by courtesy of John Wiley and Sons).

**Figure 4 materials-16-01123-f004:**
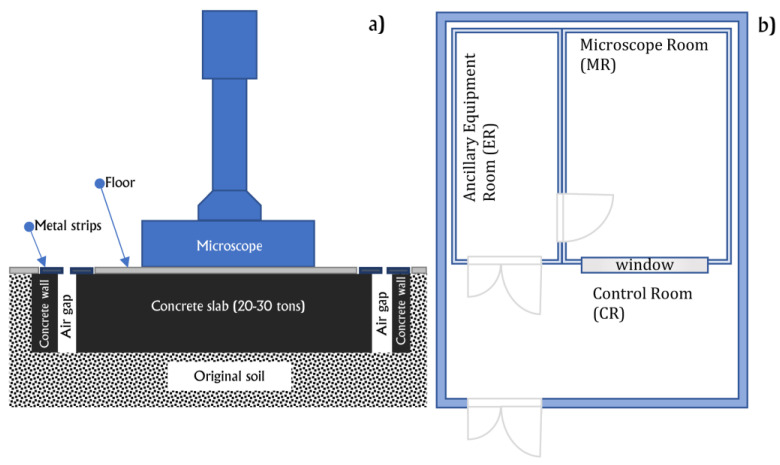
(**a**) Scheme of the anti-vibration block under the microscope in the Instrument Room (not in scale); (**b**) Scheme describing the room-in-room concept design.

**Table 1 materials-16-01123-t001:** Transmission electron microscopy facilities and relevant instrumentation, of which laboratory design and construction are discussed in the literature in dedicated articles or sections of articles. Here, details about the design of both building and laboratory hosting advanced TEM/STEM instrumentation are synoptically reported, together with the main characteristics of the instrumentation.

Facility (Name/Institution and Location)	Building Construction Details	Laboratory Design	Microscope Name (Year of Installation) (Microscope Info: Maximum Voltage, Electron Optics Peculiarities, Reached Resolution)	Additional Data on Microscope Performances and Noise Shielding
**Brookhaven National Laboratory** Long Island, NY 11973, USA [39,40]	Completely renewed 50-year-old building (previously a gym)	*no details are given*	**JEOL JEM2200FS (2004, Jeol Ltd. Akishima, Tokio, Japan)** (200 kV Schottky FEG, probe corrected, in column energy filter, 0.12 nm information limit, HAADF STEM resolution 0.105 nm) **JEOL JEM2200MCO (2008)** (200 kV Schottky FEG, URP objective lens, monochromator, double corrected, in column energy filter, 0.1 nm point and HAADF resolution, energy resolution of the omega filter 1.0–1.1 eV at ~100 µA emission current and 0.7 eV at ~30 µA)	→ Both microscope columns based on JEM2010F design, with 25 cm diameter column. Not suitable for long corrected column (JEM2010F length 2.5 m against 3.68 m of JEM2200MCO) → Contrast dip between dumbbells in Si [110] zone axis better than 20% (for JEM2200MCO)
	→ New building on a selected 5300-acre site with few sources of vibration and EM interference → Entire building constructed on compacted structural fill, compressed to 98% maximum dry density	→ Room-in-room concept: Instrument Room (IR), Equipment Room (ER) 15 cm air gap between inner and outer walls. External room with walls and ceiling in aluminum prefabricated modules, internally covered with 10-cm thick polyurethane foam insulation panels. → Control Room (CR) with double-glass panels for viewing the microscope room → Separated 3-m galley for all vibrating equipment (vacuum pumps and water chillers)	**Hitachi HD2700C (2007)** (200 kV cold FEG, dedicated STEM, probe corrected, 0.1 nm HAADF resolution, 0.35 eV energy resolution)	→ Equipped with a telephone-booth-like metal box to reduce acoustic noise and thermal drift → 24 cm column diameter → 56% contrast between Ba and background in HAADF image of BaTiO_3_
			**Titan 80-300 (2007)** (300 kV Schottky FEG, image Cs corrector, environmental TEM, 0.07–0.08 nm information limit, 0.66 eV energy resolution at 300 kV)	→ 30 cm column diameter specifically designed for mechanical and thermal stability → Contrast dip between dumbbells in Si [211] zone axis of about 20%
**Max-Planck-Institut für Metallforschung** Stuttgart, GERMANY [30]	Newly designed and constructed room		**JEM ARM 1250 (1994)** (1250 kV, thermionic LaB_6_ cathode, 0.105 nm point resolution, 0.085 nm information limit)	→ High voltage stability: <10^−6^/min p-p → Objective lens current stability <6 × 10^−7^/min p-p → Specimen drift ≤ 0.004 nm/min → ΔE: 0.6–1.6 eV depending on the operation and acquisition conditions
**One-Ångström Microscope Lab (OÅM)** **Lawrence Berkeley National Laboratory** Berkeley, USA [20,31,35,41,42]	Newly designed and constructed building	→ IR and separated ER for all noisy ancillary equipment. Walls between IRs and back rooms up to the base of the second floor to ensure acoustic separation.	**Philips CM300UTFEG (2001)** (300 kV, Schottky FEG, HREM resolution 0.089 nm, 0.078–0.080 nm information limit, 0.85 eV gun energy spread)	→ Improved information limit from 0.107 to 0.078 thanks to the high-stability of the power supplies, and hardware corrector for three-fold astigmatism → Sub-Å resolution can be accessed (in the absence of a TEM Cs-corrector) using the focal-series reconstruction (FSR) technique
**The Triebenberg Laboratory** Dresden, GERMANY [20,33]	Laboratory designed and constructed from the outset on a site selected ad hoc for its peculiarities of isolation and distance from populated areas	→ Two buildings, one for media supply, power control and conditioning system, the other for microscopes → Microscope building with six microscope units, each consisting of a microscope room, a room for peripheral devices (power supply, computers, cooling units), and an office → Room-in-room design with the interior walls of the IR 36-cm thick, at 10-cm separation from the external walls, and on a separate foundation	**Philips CM30FEG UT/Special-Tübingen TEM (2000)** (200 kV Schottky FEG, point resolution 0.165 nm (5.9 nm^−1^), information limit 0.091 nm (11 nm^−1^))	The spatial resolution of the CM30FEG improved from 1.2 Å to 0.9 Å when re-sited in the Triebenberg Laboratory
**Advanced Microscopy Laboratory** **Oak Ridge National Laboratory** Oak Ridge, Tennessee, USA [20,36,37,41]	→ New specially designed building → Building with “house-in-house” design. External walls with 12-inch concrete blocks and internal room walls with 8-inch concrete blocks → “Slab-on-grade” foundation, with instrument room slabs and wall footings on a previously prepared site comprising several layers of “engineered fill” (to a depth of 8 feet) separated by layers of a “geotechnical fabric” material that together provide a stable, uniform base for the laboratory	→ IRs separated from CR and sharing an acoustically isolated common chase, for all ancillary equipment, except water chillers → IR floor slabs (1′ thick, and the full area of the room) isolated from the CRs, corridors and service chase Access to CR through a vestibule and an air lock access slot (space) → Isolated mechanical building (200 feet from the microscope suite) for dedicated 75 kVA power supply unit, air handling systems, water chiller units, each supported on separate slabs → Separate control of airflow and temperature for each area	**JEOL JEM 2200FS (2004)** (200 kV Schottky FEG, probe Cs corrected, in column energy filter, information limit 0.085 nm, energy spread from 1.3 eV down to 0.7 eV depending on the gun conditions) **VG HB-501 (2004)** (Dedicated probe Cs corrected STEM) **VG HB-603UX (2004)** (Dedicated probe Cs corrected STEM, 0.05 nm nominal resolution)	(Data relevant to JEOL JEM 2200 FS) → Operated solely via remote computer control, no standard viewing chamber with fluorescent screen provided → Measured HT voltage stability of 0.6 × 10^−6^ (rms) and OL current stability of 0.25 × 10^−6^ (rms) giving a defocus spread of 1.85 nm and an information limit of 0.085 nm → Just after the installation, due to bad environment conditions, scarcely resolved dumbbell spacings of 0.136 nm in Si [110] similar to the same instrument without Cs aberration corrector

**Table 2 materials-16-01123-t002:** Description of measures and strategies to mitigate the noise adopted in the electron microscopy facilities considered in Table 1.

			Noise Sources, Limits and Reduction Measures				
Institution/Facility Instrumentation			EM Fields (mG, rms^1^ Values for AC Fields)	Mechanical Vibrations from Soil (Amplitude in µm (p-p) ^1^, or Velocity in µm/s)	Mechanical Vibrations from Air (Airflow: m/s)	Acoustic Noise (dB)	Temperature (t: °C), Thermal Stability (s: °C/h), Humidity (h: %)
**Brookhaven National Laboratory** Long Island, NY, 11973 USA							
	**JEM2200FS TEM/STEM** **JEM2200MCO TEM/STEM**	**Factory Limits**	**<0.5 mG** at 60 Hz	*Not reported*	**<7.6 × 10^−2^ m/s**	*Not reported*	s: **0.1 °C/h**
		**Measures** **adopted**	EM cancellation system	→ 60-cm thick (2 ft) concrete slab, isolation gap filled with de-coupling materials → Active compensation system	→ U-shaped air-supply inlet tube covered with a small pored “duct sock” → Clamshell for sample stage	*Not reported*	*Not reported*
		**Issues**	→ The system can only cancel the field at one point → Non effective for small corrections (reached values 0.2–0.5 mG at 60 Hz)	Active compensation is not suitable for frequencies lower than 10 Hz	The 7.6 × 10^−2^ m/s limit is too weak for tall instruments with aberration correctors. More stringent limits are required	*Not reported*	*Not reported*

	**Hitachi HD2700C** **Titan 80-300**	**Factory Limits**	→ AC fields **<0.035 mG** at *f* = 60 Hz **<0.035 × (*f*/60) mG** at *f* < 60 Hz → DC fields **<1 mG** (vertical) **<0.01 mG** above earth ambient field (horizontal)	**<0.25 μm/s** (rms, for all directions and frequencies)	**<1.7 × 10^−4^ m/s** (vertically) **0 m/s** (horizontally)	**<40 dB**	t: **21.1 °C** s: **0.1 °C/h** h: **40–60%**
		**Measures** **adopted**	→ EM shielding of the building electrical room by Al and low-carbon steel plates → Dimmable incandescent lighting to eliminate radio-frequency interference → Conductive and grounded floor tiles to avoid electrostatic charges → All circuits enclosed in metal conduits, electrical panels with Al and steel shielding	→ 60-cm thick concrete slab with 15 cm thick top layer containing a vibration-reducing agent “Concredamp” reinforced with polypropylene fibers → 1.3 cm isolation gap between the slab and the remaining floor → Three active vibration dampers → All vibrating equipment, such as vacuum pumps and water chillers, in a separate galley	→ Acoustic blankets above the microscope’s column to blank off air flow → Ventilation to CR only by exhaust grill located at floor level at 7.6 × 10^−2^ m/s	→ Insulating polyurethane foam panels on the outer room walls and ceiling → Silencers installed in the air handlers of the ER conditioning system → Water flow below 0.9 m/s for piping and 0.6 m/s for radiant panels → Suitable hole size in the ceiling	→ Radiant panels on the wall and ceiling in the IR → ER conditioned with constant volume VAV box and thermally insulated with gasketed doors
		**Issues**	*Not reported*	*Not reported*	Residual noise at 4–10 Hz due to belt-driven equipment	*Not reported*	*Not reported*

**Max-Planck-Institut für Metallforschung** Stuttgart, GERMANY	**JEOL JEM ARM 1250**	**Factory Limits**	AC fields **<1 mG** *	**<1 µm** (rms) at resonance	**<0.1 m/s**	*Not reported*	s: ±1 °C/h s (cooling water): **<0.05 °C/min**
		**Measures** **adopted**	*Not reported*	215 tons concrete foundation suspended by pneumatic vibration isolators (resonance frequency below 1 Hz)	*Not reported*	*Not reported*	*Not reported*
		**Issues**	*Not reported*	*Not reported*	*Not reported*	*Not reported*	*Not reported*
**One-Ångström Microscope (OÅM) Lab** **Lawrence Berkeley National Laboratory** Berkeley, USA							
	**Philips CM300UTFEG**	**Factory Limits**	AC fields **<0.1 mG** * at 60 Hz	**0.8 µm/s** at 1–5 Hz **6 µm/s** above 10 Hz (horizontal, left to right)	*Not reported*	*Not reported*	s: **0.5 °C/h**
		**Measures** **adopted**	→ Power and signal cables, and all cooling-water hoses, routed in steel-covered cable trenches far from the microscope	→ Concrete isolation slab (3.3 m × 4.2 m, 1 m thick) with 2.5 cm isolation gap (vibration reduced by a factor three/four vertically, also at 1–5 Hz, and more than 10 times in the other directions)	→ Air inlets along the side of the room, farthest from the microscope column, providing a laminar flow down the wall and across the floor	→ Acoustic damping by 50-mm thick cloth-covered fiberglass sound absorbent on both sides of the wall separating the ER from the IR → All noisy equipment (vacuum pumps, water chillers, HT tank and computers) in a separate ER. Solid-state amplifiers to extend keyboard, mouse, and monitor cables to 7.5 m. Microscope camera controllers moved from the microscope console to the ER and covered with acoustic panels → Carpet over thick rubber pad on the second floor to mitigate foot fall impacts	Water chiller for objective lens coil adjusted so that the temperature of the water leaving the lens is at the temperature of the microscope room
		**Issues**	*Not reported*	*Not reported*	*Not reported*	*Not reported*	*Not reported*
**The Triebenberg Laboratory** Dresden—GERMANY							
		**Factory** **Limits**	AC fields **<0.05 mG * at 60 Hz.** (Note: Before microscope installation AC stray fields were 2 µG)	*Not reported*	**0.05 m/s**	**<20 dB**	**s < 0.1 °C /min**
		**Measures** **adopted**	→ Transformer at 100 m from the microscope and suitably oriented for minimal stray fields → All cables in the laboratory twisted and shielded for short range damping of the stray fields → Earth connection at one point without any ground loop No gas discharge lights → Only flat panel display for computers	→ Entire Building on a 2-m thick layer of sand → Three mutually separated foundations for the outer building, inner building and concrete slab for the microscope → Walls of the microscope building with high density material	Air-inlet through hollow floor, optimized by computer simulation	→ Air ducts covered by 2-cm thick, porous rubber → Acoustic damping systems applied to all ventilation units → No devices cooled by air-blowers admitted in the microscope rooms	The room heat capacity allows to switch off the air conditioning systems during critical experiments
		**Issues**	*Not reported*	*Not reported*	*Not reported*	*Not reported*	*Not reported*
**Advanced microscope laboratory (AML)** **Oak Ridge National Laboratory** Oak Ridge, TN, USA							
	**Jeol 2200FS** **VG HB501 and VG HB 603UX**	**Factory Limits**	AC fields **<0.05 mG * at 60 Hz**	**<1 µm/s** at 1–30 Hz			**0.2 °C/h**
		**Measures** **adopted**	→ In the foundation, epoxy-coated re-bars tied together with plastic-coated wire to minimize the possibility of magnetic fields caused by induced currents → Dielectric decoupling units installed every 10 to 15 feet in all water lines, metal air ducts, compressed air lines and fire sprinkler piping to avoid field generation by currents carried in other laboratory systems → Twisted-pair wiring throughout both the instrument and mechanical buildings	*Not reported*	Cooling air entrance in the IR through a pair of large, perforated supply ducts (50% open area) into a 1.5-m high volume above a porous acoustic ceiling, providing a downward flow to the floor into plenums on two side walls	→ In the IR, special acoustic/absorber blankets on the walls to dampen any noise generated in adjacent rooms → Cloth-covered acoustic absorber panels on each wall of the CR to absorb noise from conversation and computer fans	*Not reported*
		**Issues**	*Not reported*	*Not reported*	*Not reported*	→ 800 Hz noise due to acoustic coupling between TMP and column, attenuated by moving the TMPs far from the column and acoustically insulating them	Not reported
		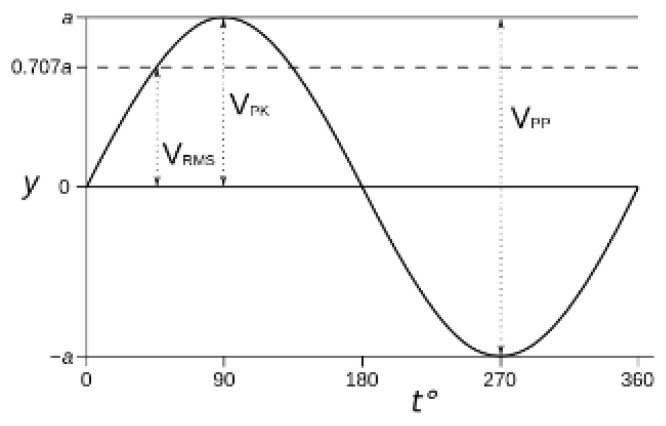		^1^ rms = root-mean-square, p-p = peak-to-peak; p-p = (2$\sqrt{2}$ rms) for sinusoidal waves. For EM fields, all p-p values are converted into rms. When * is placed close to the number, it means that it is not specified in the relevant paper if the value is p-p or rms. Since the effective value of an AC field is the rms one, it is reasonable to assume that when not specified, the given value is the rms one.			

## Data Availability

All the data available are contained in this paper.

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
