# Peer review of "The Relevance of Building an Appropriate Environment around an Atomic Resolution Transmission Electron Microscope as Prerequisite for Reliable Quantitative Experiments: It Should Be Obvious, but It Is a Subtle Never-Ending Story!"

_materials, 2023, doi:10.3390/ma16031123_

Round 1

Reviewer 1 Report

The Review paper “The relevance of building an appropriate environment around an atomic resolution transmission electron microscope as prerequisite for reliable quantitative imaging and spectroscopy experiments: It should be obvious, but it is a subtle never-ending story!” it’s an interesting study which wants to highlight the state-of-the art for the realization of laboratory environment for the modern TEM/STEM’s.

Unfortunately, I don’t find it relevant because the realization of the environment for TEM’s it is a particular case for each TEM (depending of each system, with/without corrector/monochromator, what kind of cameras/detector are chosen, etc.), so different environmental conditions, we can't compare. Anyway, nowadays, the interaction between the laboratory which want to buy a microscope and the company who build the microscope it’s very close and they will provide all the data for the environment construction.

The modern microscopes are expensive, of course, but the issues that can appear during the installation or after are different from case to case and can be easily resolved, being covered by the service contract.  

The paper it’s not suitable for the publication in Materials Journal because it's not a scientific research.

Author Response

We would like to thank the referee #1 for reading our work. His way of thinking is precisely the reason why we decided to spend our efforts to write this paper.

Reviewer 2 Report

In the presented review the authors describe the requirements on the room design and infrastructure and suggest some optimizing procedures based on their vast experience in electron microscopy and examples from literature. With their article they address a highly relevant topic which, very often, receives to little attention especially outside of the microscopy community.

The manuscript is interesting to read and presents some good examples. Especially the list of installations and laboratory designs I found particularly useful. There are only a few minor issues to mention:

- Generally, the authors should improve the language a bit: There are quite a few typos and inappropriate/overcomplicated choice of words (for instance: page 3: “…require the same skillful” probably means “skill set” or “skills”; Page 3: ”because they do are aware” should be “because they are aware”; page 3: “without no interaction”  should be “without interaction”, Page 4: “cannot do understand” should be “cannot understand”, page 6: “order of reasons” means simply “reasons”, …). Also, in some cases, the sentences are unnecessarily complicated with some degree of circumlocution, consider rephrasing.

- On page 6: I do not completely agree with the very general statement that STEM techniques suffer more from instabilities, due to longer acquisition times compared to TEM, because this is highly dependent on the experimental settings and the method/contrast mechanism used. Maybe the authors can specify and elaborate on this statement and provide some examples and references? However, it is true that the main difference lies in the way how instabilities become apparent in the data. Therefore, it is often “much easier” to correct for scan distortions by applying correction algorithms to the data (as shown for instance for image data by Lewys Jones et al., DOI: 10.1017/S1431927613001402.).

- Page 19: “one thousand higher than 40 dB”, it should be “increased by a factor of one thousand…”

- Also consider shortening the title which is a bit long

A believe that the manuscript gives a good overview over possible hurdles and things to consider while designing a (S)TEM – room. However, the clarity of the otherwise very good manuscript would profit a lot from the suggested improvements. Apart from these minor issues I suggest publication.

Author Response

Answer to the Referee #2:

We would like to thank the referee #2 for his positive evaluation and for the suggestions to improve further our work. In the following the point-to-point answers and the relevant amendments marked in yellow in the paper:

Q: - Generally, the authors should improve the language a bit: There are quite a few typos and inappropriate/overcomplicated choice of words (for instance: page 3: “…require the same skillful” probably means “skill set” or “skills”; Page 3: ”because they do are aware” should be “because they are aware”; page 3: “without no interaction” should be “without interaction”, Page 4: “cannot do understand” should be “cannot understand”, page 6: “order of reasons” means simply “reasons”, …). Also, in some cases, the sentences are unnecessarily complicated with some degree of circumlocution, consider rephrasing

A: -       “Skillful” has been replaced by “Skill set”.

  • ”because they do are aware” has been replaced by “because they are aware”.
  • “without no interaction” has been replaced by “without interaction”.
  • “cannot do understand” has been replaced by “cannot understand”.
  • “order of reasons” has been replaced by “reasons”,
  • Rephrasing has been considered and the further amendments are marked in yellow in the paper.

Q: - On page 6: I do not completely agree with the very general statement that STEM techniques suffer more from instabilities, due to longer acquisition times compared to TEM, because this is highly dependent on the experimental settings and the method/contrast mechanism used. Maybe the authors can specify and elaborate on this statement and provide some examples and references? However, it is true that the main difference lies in the way how instabilities become apparent in the data. Therefore, it is often “much easier” to correct for scan distortions by applying correction algorithms to the data (as shown for instance for image data by Lewys Jones et al., DOI: 10.1017/S1431927613001402.).

A: – The sentence has been modified as follow also adding the relevant references:

“During sequential acquisition, these effects typically consist in image drifts and distortions, which required the development of dedicated corrections algorithms [55] whereas, during parallel acquisition, loss of contrast and, therefore, of information, is observed [27, 34, 37, 39]. The recent introduction of high speed, high sensitivity direct conversion detectors [56,57] improved STEM imaging SNR very much, while reducing the acquisition time, but at the same time made possible the development of a new powerful approach, named 4D-STEM, requiring even longer acquisition time, dedicated noise reduction algorithms, and much more care in the laboratory stability design [58,59].” 

Q: - Page 19: “one thousand higher than 40 dB”, it should be “increased by a factor of one

thousand…”

A: - - Page 19: “one thousand higher than 40 dB”, has been replaced by “increased by a factor of one thousand…”.

Q: - Also consider shortening the title which is a bit long

A: - the title has been shortened a bit.

Reviewer 3 Report

In this review, the authors summarized the source of environmental noise and provide advice and experience to eliminate those noises. It shows the importance of designing, constructing, and maintaining an appropriate environment around atomic resolution transmission electron microscopes. Such information is of great importance yet often overlooked by non-experts in the relevant or similar field. I personally found the contents very informative and acknowledge authors’ effort to put them together. Therefore, I highly support its publication in Materials despite there are a few editorial errors.

a few citation errors on page 4 and 24.

The tables can be better organized.

Author Response

Answer to the Referee #3:

We would like to thank the referee #3 for his positive evaluation and for the suggestions to improve further our work. In the following the point-to-point answers and the relevant amendments marked in yellow in the paper:

Q:- a few citation errors on page 4 and 24.

A:- The typos citation error for the references on page 20 and 24 have been amended

Q:- The tables can be better organized

A: - The tables reported in the paper are the best results of several improving attempts. Following your suggestion, we tried to improve them further but without convincing results. Nevertheless, we believe that these tables could, even in the present form, be useful for the readers.

Reviewer 4 Report

This manuscript is an overview of the technical issues to deal with when designing and building the infrastructure to accommodate high-resolution TEM/STEM instruments. The importance of different factors affecting not only the space resolution, but also the signal-to-noise and signal-to-background ratios of the analytical TEM/STEM instruments are presented, among which the electromagnetic noise, the thermal and mechanical stability. To illustrate the relevance of the analyzed factors, two extensive tables are provided containing technical solutions adopted by worldwide reputed TEM laboratories when installing or relocating high-resolution TEM/STEM instruments in new, specially designed buildings or rooms, in function of the TEM/STEM manufacturer requirements and local environmental conditions. Keeping in mind the high costs of the advanced TEM/STEM nowadays corelated with their scientific potential, this table will be extremely useful for the people situated in deciding positions as well as for those directly involved in preparing the infrastructure for properly hosting TEM/STEM instruments in order to profit from the highest technical specifications of the instrument.

The manuscript clearly bears the mark of some negative experience accumulated by the authors in the field of designing the appropriate infrastructure for advanced TEM equipment, representing also a strong advice and call to all the people involved in the complex effort of building a TEM/STEM laboratory to take into consideration the signaled technical issues as well as the opinion and expertise of experienced TEM/STEM specialists who have successfully designed such facilities, as demonstrated by the technical performances reached by the instruments installed in optimal conditions.   

Apart from a few minor editing issues that will be most probably addressed on proofing, such as:

-          Page 2, 3-rd row from the bottom: “worsen” suggested instead of “worst”

-          Page 3, 1-st row: “do realize”, “are indeed aware”, “do understand” suggested instead of “do are aware”

-          Page 4, 3-rd row: “we cannot really understand” suggested instead of “we cannot do understand”

-          Page 4, 4-th row of section “2. Historical survey”: the mention “Error! Bookmark not defined” should be removed

-          Page 6, title of section 3: “noise” suggested instead of “noises”

-          Page 18, 1-st row of section “3.4 Sources of mechanical noise”: “very” suggested to be removed

-          Page 20, 3-rd row: “lets” suggested instead of “let”

-          Page 24, : the mention “Error! Bookmark not defined” should be removed

I recommend the publication of this manuscript in its present form.

Author Response

Answer to the Referee #4:

We would like to thank the referee #4 for his positive evaluation and for the suggestions to improve further our work. In the following the point-to-point answers and the relevant amendments marked in yellow in the paper:

Q:- Page 2, 3-rd row from the bottom: “worsen” suggested instead of “worst”

A:- “worst” has been replaced by “worse”

Q:- Page 3, 1-st row: “do realize”, “are indeed aware”, “do understand” suggested instead of “do are aware”

A:- The statement has been modified and it is marked in yellow in the text.

Q:- Page 4, 3-rd row: “we cannot really understand” suggested instead of “we cannot do understand

A:- The statement has been modified and it is marked in yellow in the text.

Q:- Page 4, 4-th row of section “2. Historical survey”: the mention “Error! Bookmark not defined” should be removed

A:- It has been amended

Q:- Page 6, title of section 3: “noise” suggested instead of “noises”

A:- It has been modified accordingly

Q:- Page 18, 1-st row of section “3.4 Sources of mechanical noise”: “very” suggested to be removed

A:- It has been modified accordingly

Q:- Page 20, 3-rd row: “lets” suggested instead of “let”

A:- we did not find this typos.

Q:- Page 24, : the mention “Error! Bookmark not defined” should be removed

A:- It has been amended
